# Occurrence of the megatoothed sharks (Lamniformes: Otodontidae) in Alabama, USA

Dana J. Ehret[1] and Jun Ebersole[2]

[1] Alabama Museum of Natural History, Tuscaloosa, AL, USA
[2] McWane Science Center, Birmingham, AL, USA

## ABSTRACT

The Otodontidae include some of the largest sharks to ever live in the world's oceans (i.e., *Carcharocles megalodon*). Here we report on Paleocene and Eocene occurrences of *Otodus obliquus* and *Carcharocles auriculatus* from Alabama, USA. Teeth of *Otodus* are rarely encountered in the Gulf Coastal Plain and this report is one of the first records for Alabama. *Carcharocles auriculatus* is more common in the Eocene deposits of Alabama, but its occurrence has been largely overlooked in the literature. We also refute the occurrence of the Oligocene *Carcharocles angustidens* in the state. Raised awareness and increased collecting of under-sampled geologic formations in Alabama will likely increase sample sizes of *O. obliquus* and *C. auriculatus* and also might unearth other otodontids, such as *C. megalodon* and *C. chubutensis*.

## INTRODUCTION

The megatoothed sharks (Family Otodontidae) are well known in the marine fossil record of the Paleocene through Pliocene. These large, macro-predatory sharks are cosmopolitan in their distributions, and they are present in the fossil records of Asia, Africa, Europe, and North and South America (*Cappetta, 2012*). Beginning with *Otodus obliquus* (*Agassiz, 1838*) in the Paleocene and including the largest shark that ever lived, *Carcharocles megalodon* (*Agassiz, 1835*), otodontids are arguably the most well known of all fossil chondrichthyans. While *C. megalodon* is probably the most abundant and widely recognized species, other species of *Otodus* and *Carcharocles* are less often reported in the literature, which may have had a negative affect on the distribution and abundance of these species (*Cappetta, 2012*). This discrepancy might be related to a sampling bias, the familiarity of *C. megalodon* compared with other megatoothed species, or it might actually reflect the dispersal patterns of these other otodontid species.

Previous reports of otodontids from Alabama have varied in accurate identifications, including references to *C. auriculatus* (*White, 1956*; *Thurmond & Jones, 1981*), *C. angustidens* (*Agassiz, 1835*), *Otodus crassa* (*Agassiz, 1843*), and *O. crassus* (*Gibbes, 1848*). Although middle Eocene outcrops are fairly prevalent in Alabama, studies of the otodontids have largely been overlooked in the state. Here we present and discuss records of *O. obliquus* and *C. auriculatus* (*Blainville, 1818*) in the Paleocene and Eocene of Alabama, respectively.

Corresponding author
Dana J. Ehret, djehret@ua.edu

## MATERIAL AND METHODS

The collections of the Alabama Museum of Natural History (ALMNH) in Tuscaloosa, the Geological Survey of Alabama (GSA) in Tuscaloosa, and McWane Science Center (MSC/RMM) in Birmingham were examined for specimens of otodontid sharks from Alabama. All three collections contained specimens that are previously unreported in the literature. In these collections, most specimens of *C. auriculatus* were correctly identified, however many of the *Otodus* specimens were incorrectly assigned to either *Carcharias* or *Lamna*. These misidentifications are likely the reason that *Otodus* has not been accurately reported from the state previously.

Five *O. obliquus* specimens were identified in the collections of the Geological Survey of Alabama (GSA) (Table 1). These specimens were collected in the late 1800s and early 1900s, with all being unidentified or misidentified. Specimens of *Carcharocles auriculatus* located in the ALMNH and MSC collections were collected over the last century from Choctaw, Clarke, Covington, Washington, and Wilcox counties (Fig. 1 and Table 1). All *C. auriculatus* specimens were collected in Early to Middle Eocene deposits (mainly Lutetian and Bartonian) of southwestern Alabama and all specimens examined in this study were found by surface collection methods over the past 100+ years.

### Geologic setting

In Alabama, *Otodus obliquus* and *Carcharocles auriculatus* specimens have been collected from lithostratigraphic units ranging from the early Paleocene to Middle Eocene including the Midway, Sabine, Claiborne, Jackson groups (Figs. 1, 2 and Table 1). The Paleocene and Eocene formations in the state make up a nearly time-continuous series that ranges from the K/Pg contact to the Eocene/Oligocene contact (*Raymond et al., 1988*). A small unconformity exists between the upper-most Cretaceous units in the state, the Prairie Bluff Chalk and Providence Sand, and the lower-most Paleocene Clayton Formation. The Clayton Formation (which includes the Pine Barren and McBryde Limestone members) is the basal unit in the Midway Group, a group that also includes, in ascending order, the Porters Creek (with the Matthews Landing Marl Member) and Naheola (with the Oak Hill and Coal Bluff Marl members) formations. The Midway Group is conformably overlain by the Paleocene/Eocene Wilcox Group. The Paleocene units within the Wilcox Group include, in ascending order, the Nanafalia Formation (with the Gravel Creek Sand Member, an informal unit referred locally as the "*Ostrea thirsae* beds", and the Grampian Hills Member), and the Tuscahoma Sand (which includes the Greggs Landing Marl and the Bells Landing Marl members).

The uppermost unit of the Wilcox Group is the Early Eocene (Ypresian) Hatchetigbee Formation, which contains the Bashi Marl Member at its base. The Wilcox Group is disconformably overlain by the lithostratigraphic units within the Claiborne and Jackson groups. The Claiborne Group consists of, in ascending order, the Tallahatta and Lisbon (with informal "lower", "middle", and "upper" members) formations, and the Gosport Sand. The Jackson Group includes the Moodys Branch and Crystal River formations and the Yazoo Clay. The Yazoo Clay in Alabama is further subdivided into the following

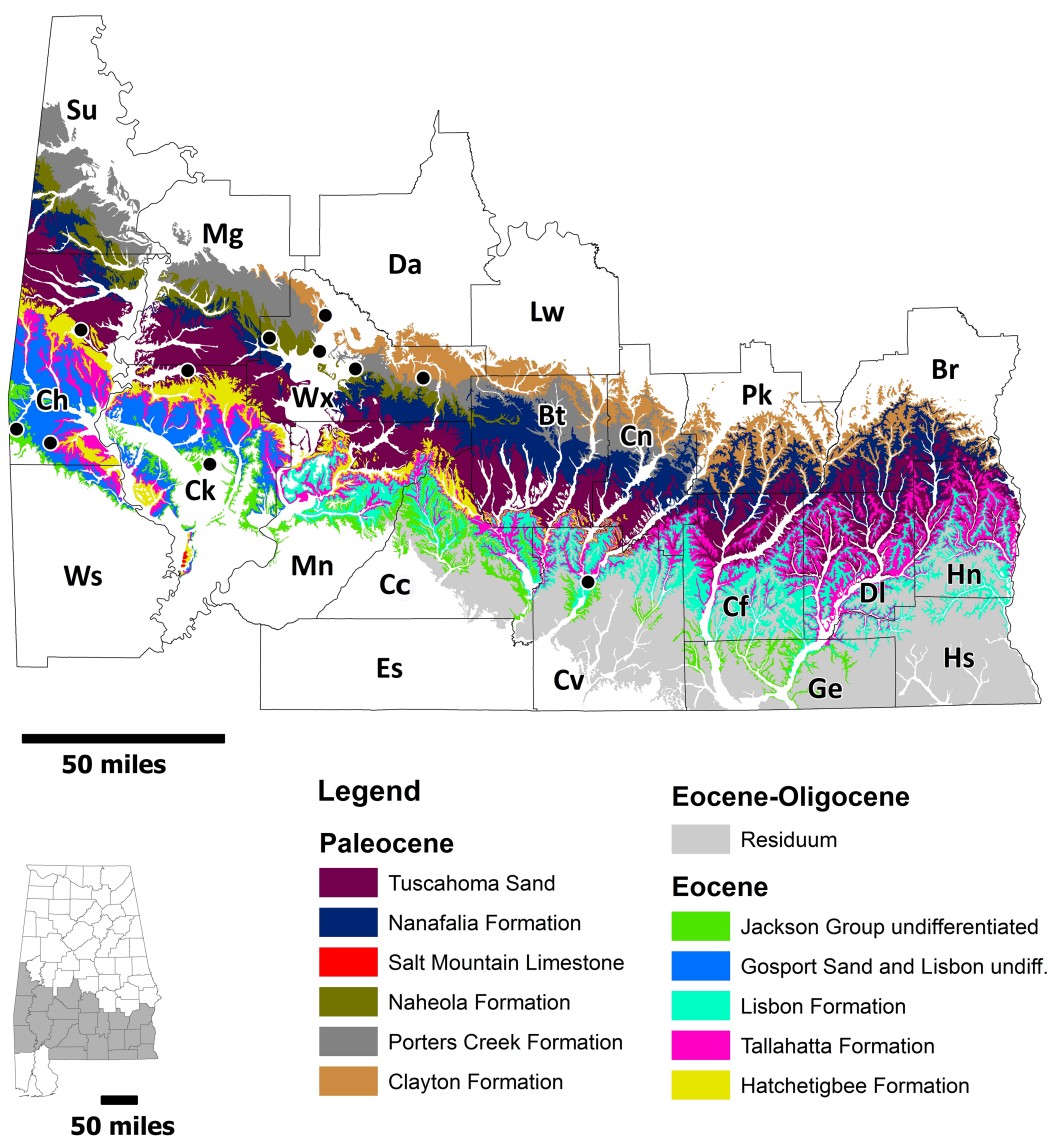

**Figure 1 Geologic map of Paleocene and Eocene strata in Alabama.** Map showing Paleocene and Eocene strata in Alabama and known collecting localities for otodontid specimens. County abbreviations: Br, Barbour; Bt, Butler; Cc, Conecuh; Cf, Coffee; Ch, Choctaw; Ck, Clarke; Cn, Crenshaw; Cv, Covington; Da, Dallas; Dl, Dale; Es, Escambia; Ge, Greene; Hn, Henry; Hs, Houston; Lw, Lowndes; Mg, Marengo; Mn, Monroe; Pk, Pike; Su, Sumter; Ws, Washington, and Wx, Wilcox. Compiled using Geological Survey of Alabama digital geology data (*GSA , 2006*, adapted from *Szabo et al., 1988*).

**Table 1 Otodus and Carcharocles specimens in museum collections from Alabama.** *Otodus* and *Carcharocles* specimens from the Alabama Museum of Natural History (ALMNH), Geological Survey of Alabama (GSA), and McWane Science Center collections (MSC/RMM).

| Catalog/accession number | Genus | Species | Formation/unit | Stage | County | State | Locality |
|---|---|---|---|---|---|---|---|
| ALMNH 1985.30.2 | *Carcharocles* | *auriculatus* | Unknown | Unknown | Unknown | AL | Unknown |
| ALMNH 1985.35.2 | *Carcharocles* | *auriculatus* | Unknown | Unknown | Unknown | AL | Unknown |
| ALMNH 1985.72.28.5 | *Carcharocles* | *auriculatus* | Hatchetigbee or Lisbon Fm. | Ypresian or Lutetian/Bartonian | Choctaw | AL | Shell Creek |
| ALMNH 1985.72.33 | *Carcharocles* | *auriculatus* | Unknown | Unknown | Unknown | AL | Unknown |
| ALMNH 1985.72.43.3 | *Carcharocles* | *auriculatus* | Unknown | Unknown | Unknown | AL | Unknown |
| ALMNH 1985.72.55.2 | *Carcharocles* | *auriculatus* | Unknown | Unknown | Unknown | AL | Unknown |
| ALMNH 1985.72.62.3 | *Carcharocles* | *auriculatus* | Unknown | Unknown | Unknown | AL | Unknown |
| ALMNH 1985.72.83 | *Carcharocles* | *auriculatus* | Unknown | Unknown | Unknown | AL | Unknown |
| ALMNH 1985.72.84 | *Carcharocles* | *auriculatus* | Unknown | Unknown | Unknown | AL | Unknown |
| ALMNH 1985.72.88 | *Carcharocles* | *auriculatus* | Unknown | Unknown | Unknown | AL | Unknown |
| ALMNH 1988.1.9 | *Carcharocles* | *auriculatus* | Yazoo Clay | Bartonian/Priabonian | Unknown | AL | Unknown |
| ALMNH 1988.29.1 | *Carcharocles* | *auriculatus* | Gosport Sand | Bartonian | Choctaw | AL | Puss Cuss Creek |
| ALMNH 1989.4.50.1 | *Carcharocles* | *auriculatus* | Lisbon Fm. | Lutetian/Bartonian | Choctaw | AL | Butler |
| ALMNH 1992.28.44.1 | *Carcharocles* | *auriculatus* | Lisbon–Tallahatta Contact | Lutetian | Covington | AL | Point-A Dam |
| ALMNH 1992.28.44.2 | *Carcharocles* | *auriculatus* | Lisbon–Tallahatta Contact | Lutetian | Covington | AL | Point-A Dam |
| ALMNH 2000.1.4.1 | *Carcharocles* | *auriculatus* | Yazoo Clay | Bartonian/Priabonian | Choctaw | AL | Unknown |
| ALMNH 2000.1.16.1 | *Carcharocles* | *auriculatus* | Yazoo Clay | Bartonian/Priabonian | Choctaw | AL | Unknown |
| ALMNH 2000.1.27.1 | *Carcharocles* | *auriculatus* | Yazoo Clay | Bartonian/Priabonian | Choctaw | AL | Unknown |
| ALMNH 2000.1.29.1 | *Carcharocles* | *auriculatus* | Yazoo Clay | Bartonian/Priabonian | Choctaw | AL | Unknown |
| ALMNH 2000.1.33.1 | *Carcharocles* | *auriculatus* | Pachuta Marl Member | Priabonian | Washington | AL | Bashi |
| ALMNH 2000.1.53 | *Carcharocles* | *auriculatus* | Yazoo | Bartonian/Priabonian | Choctaw | AL | Unknown |
| ALMNH 2000.1.57 | *Carcharocles* | *auriculatus* | Yazoo | Bartonian/Priabonian | Choctaw | AL | Unknown |
| ALMNH 2000.1.59 | *Carcharocles* | *auriculatus* | Yazoo | Bartonian/Priabonian | Choctaw | AL | Unknown |
| ALMNH 2005.6.259 | *Carcharocles* | *auriculatus* | Yazoo Clay | Bartonian/Priabonian | Choctaw | AL | Unknown |
| ALMNH 2005.6.279 | *Carcharocles* | *auriculatus* | Yazoo Clay | Bartonian/Priabonian | Clarke | AL | Grove Hill |
| ALMNH 2005.6.294 | *Carcharocles* | *auriculatus* | Unknown | Unknown | Unknown | AL | Unknown |
| ALMNH 2005.6.334.6 | *Carcharocles* | *auriculatus* | Tallahatta Fm. | Ypresian/Lutetian | Wilcox | AL | Prairie Bluff |
| ALMNH 2005.6.407 | *Carcharocles* | *auriculatus* | Unknown | Unknown | Unknown | AL | Unknown |
| ALMNH 2005.6.408.1 | *Carcharocles* | *auriculatus* | Unknown | Unknown | Wilcox | AL | Prairie Bluff |
| ALMNH 2010.5.3 | *Carcharocles* | *auriculatus* | Unknown | Unknown | Unknown | AL | Unknown |
| GSA 5050 | *Otodus* | *obliquus* | Matthews Landing Marl Member | Selandian | Wilcox | AL | Camden |
| GSA 5051 | *Otodus* | *obliquus* | Unknown | Unknown | Wilcox | AL | Unknown |

Table 1 (*continued*)

| Catalog/ accession number | Genus | Species | Formation/unit | Stage | County | State | Locality |
|---|---|---|---|---|---|---|---|
| GSA 5052 | *Otodus* | *obliquus* | Matthews Landing Marl Member | Selandian | Wilcox | AL | Matthews Landing |
| GSA 5053 | *Otodus* | *obliquus* | Porters Creek Formation | Danian | Wilcox | AL | Graveyard Hill No. 4 |
| GSA 5054 | *Otodus* | *obliquus* | Midway Group | Danian/Selandian | Unknown | AL | Unknown |
| MSC 20969 | *Carcharocles* | *auriculatus* | Lisbon-Tallahatta Contact | Lutetian | Covington | AL | Point-A Dam |
| MSC 20970 | *Carcharocles* | *auriculatus* | Lisbon-Tallahatta Contact | Lutetian | Covington | AL | Point-A Dam |
| MSC 20971 | *Carcharocles* | *auriculatus* | Lisbon-Tallahatta Contact | Lutetian | Covington | AL | Point-A Dam |
| MSC 20972 | *Carcharocles* | *auriculatus* | Lisbon-Tallahatta Contact | Lutetian | Covington | AL | Point-A Dam |
| MSC 20973 | *Carcharocles* | *auriculatus* | Lisbon-Tallahatta Contact | Lutetian | Covington | AL | Point-A Dam |
| MSC 20974 | *Carcharocles* | *auriculatus* | Lisbon-Tallahatta Contact | Lutetian | Covington | AL | Point-A Dam |
| MSC 20975 | *Carcharocles* | *auriculatus* | Lisbon-Tallahatta Contact | Lutetian | Covington | AL | Point-A Dam |
| MSC 20976 | *Carcharocles* | *auriculatus* | Lisbon-Tallahatta Contact | Lutetian | Covington | AL | Point-A Dam |
| MSC 20977 | *Carcharocles* | *auriculatus* | Lisbon-Tallahatta Contact | Lutetian | Covington | AL | Point-A Dam |
| MSC 20978 | *Carcharocles* | *auriculatus* | Lisbon-Tallahatta Contact | Lutetian | Covington | AL | Point-A Dam |
| MSC 20979 | *Carcharocles* | *auriculatus* | Lisbon-Tallahatta Contact | Lutetian | Covington | AL | Point-A Dam |
| MSC 20980 | *Carcharocles* | *auriculatus* | Lisbon-Tallahatta Contact | Lutetian | Covington | AL | Point-A Dam |
| MSC 20981 | *Carcharocles* | *auriculatus* | Lisbon-Tallahatta Contact | Lutetian | Covington | AL | Point-A Dam |
| MSC 20982 | *Carcharocles* | *auriculatus* | Lisbon-Tallahatta Contact | Lutetian | Covington | AL | Point-A Dam |
| MSC 20983 | *Carcharocles* | *auriculatus* | Lisbon-Tallahatta Contact | Lutetian | Covington | AL | Point-A Dam |
| MSC 20984 | *Carcharocles* | *auriculatus* | Lisbon-Tallahatta Contact | Lutetian | Covington | AL | Point-A Dam |
| MSC 20985 | *Carcharocles* | *auriculatus* | Lisbon-Tallahatta Contact | Lutetian | Covington | AL | Point-A Dam |
| MSC 29068 | *Carcharocles* | *auriculatus* | Lisbon-Tallahatta Contact | Lutetian | Covington | AL | Point-A Dam |
| MSC 34422 | *Carcharocles* | *auriculatus* | Yazoo Clay - Pachuta Marl Mbr. | Priabonian | Washington | AL | Unknown |
| MSC 34423 | *Carcharocles* | *auriculatus* | Lisbon-Tallahatta Contact | Lutetian | Choctaw | AL | Silas |
| RMM 2370 | *Carcharocles* | *auriculatus* | Gosport Sand | Bartonian | Choctaw | AL | Puss Cuss Creek |
| RMM 2371 | *Carcharocles* | *auriculatus* | Gosport Sand | Bartonian | Choctaw | AL | Puss Cuss Creek |

| AGE | GRP. | ALABAMA STRATIGRAPHY (W — E) | PLANKTONIC FORAMINIFERAL ZONE | P ZONE | NP ZONE | STG. |
|---|---|---|---|---|---|---|
| EOCENE | Jackson | Yazoo Clay — Shubuta Mbr. / Pachuta Marl Mbr. / Cocoa Sand Mbr. / North Twistwood Creek Clay Member / Crystal River Fm. | *Gr. cerroazulensis* I. Z. | P 17 (in part) / P 16 | NP 20 / NP 19 | Priabonian |
| | | | *P. semiinvoluta* I. Z. | P 15 | NP 18 | |
| | | Moodys Branch Formation | *T. rohri* I. Z. | P 14 | NP 17 | Bartonian |
| | | Gosport Sand | | | | |
| | Claiborne | "upper Lisbon" / Lisbon Formation | *O. beckmanni* R. Z. | P 13 | NP 16 | |
| | | "middle Lisbon" | | | | Lutetian |
| | | "lower Lisbon" | *Gg. subconglobata* C. R. Z. | P 11 | NP 15 | |
| | | Tallahatta Formation | *H. aragonensis* I. Z. | P 10 | NP 14 / NP 13 / NP 12 | |
| | | Hatchetigbee Formation | *M. subbotinae* I. Z. | P 6 | NP 11 / NP 10 | Ypresian |
| | | Bashi Marl Member | | | | |
| PALEOCENE | Wilcox | Tuscahoma Sand — Bells Lnd. Marl Mbr. | *M. velascoensis* I. Z. | | NP 9 | |
| | | Greggs Lnd. Marl Mbr. | | P 5 | | |
| | | Nanafalia Formation — Grampian Hills Mbr. / "*Ostrea thirsae*" beds / Gravel Creek Sand Mbr. | *Pr. pseudomenardii* R. Z. | P 4 | NP 8 / NP 7 | Selandian |
| | | Naheola Formation — Coal Bluff Marl Mbr. / Oak Hill Mbr. | *Pr. pusilla pusilla* I. Z. | | NP 5 | |
| | Midway | | *M. angulata* I. Z. | P 3 | NP 4 | |
| | | Matthews Landing Marl Mbr. | | | | |
| | | Porters Creek Fm. | *M. uncinata* I. Z. | P 2 | | |
| | | Clayton Fm. — McBryde Ls. Mbr. / Pine Barren Mbr. / Clayton Formation | *S. trinidadensis* I. Z. | P 1 | NP 3 / NP 2 | Danian |
| | | | *S. pseudobulloides* I. Z. | | NP 1 | |

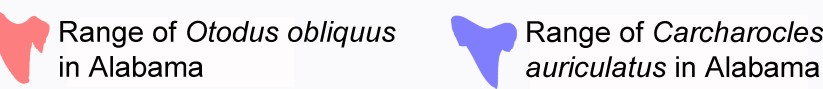

Range of *Otodus obliquus* in Alabama   Range of *Carcharocles auriculatus* in Alabama

**Figure 2  Paleocene and Eocene Stratigraphy of Alabama.** Stratigraphic chart showing the age of Paleocene and Eocene formations of Alabama. Modified from *Mancini & Tew (1991)*.

chronologic members: the North Twistwood Creek, Cocoa Sand, Pachuta Marl, and Shubuta (Fig. 2).

## SYSTEMATIC PALEONTOLOGY

Class Chondrichthyes *Huxley, 1880*
Subclass Elasmobranchii *Bonaparte, 1838*
Order Lamniformes *Berg, 1958*
Family Otodontidae *Glikman, 1964*
Genus *Otodus Agassiz, 1838*
*Otodus obliquus Agassiz, 1838*
Figs. 3A–3D and Table 1

### Referred specimens
GSA CZ 5050, GSA CZ 5051, GSA CZ 5052, GSA CZ 5053, GSA CZ 5054

**Occurrence** Wilcox County, Alabama

**Description**

*Otodus obliquus* teeth were identified using the following characteristics: triangular cusp, lacking serrations on cutting edges; labial face is moderately convex and does not overhang the root; lingual face is smooth and convex; a well developed v-shaped chevron on the lingual face; a pair of triangular cusplets that lack serrations; and a highly developed lingual protuberance of the root (*Cappetta, 2012*). Five *O. obliquus* specimens were identified in the historical collections housed at the Geological Survey of Alabama (GSA). GSA CZ 5051 (Fig. 3A) is part of the Schowalter Collection and was collected prior to 1889. Unfortunately the precise locality and formation of origin for GSA CZ 5051 is unknown as the specimen is only accompanied by a label marked "Tertiary, Wilcox", presumably referring to the Cenozoic strata in Wilcox County, Alabama. Of the Cenozoic units within this county, exposures can be found of all five Paleocene formations, which make up the Midway Group (Clayton, Porters Creek, and Naheola formations) and Wilcox Group (Nanafalia Formation and Tuscahoma Sand) in Alabama. Wilcox County also has exposures of the Ypresian Hatchetigbee Formation, also part of the Wilcox Group (Fig. 1). Based on the surface exposures of these formations, we argue this specimen is either Selandian (Naheola Formation) or Thanetian (Tuscahoma Sand) in age. The tooth is a nearly complete posterolateral that exhibits large triangular cusplets, with a secondary pair also present. GSA CZ 5051 also exhibits a v-shaped chevron on its lingual surface, and smooth cutting edges on the main cusp. Although the tip is broken, the measured main crown height is 25.5 mm, while its width is 17.9 mm.

GSA CZ 5050 (Fig. 3B) represents a right posterolateral tooth that is accompanied with a label inscribed "Sucarnoochee, Clarence Jones' Place". The term "Sucarnoochee" refers to the Sucarnoochee beds, a historical and informal unit that was described as being between the Paleocene Clayton and the Naheola formations. "Clarence Jones' Place" refers to a historic locality located near Camden in Wilcox County that is known for its fine exposures

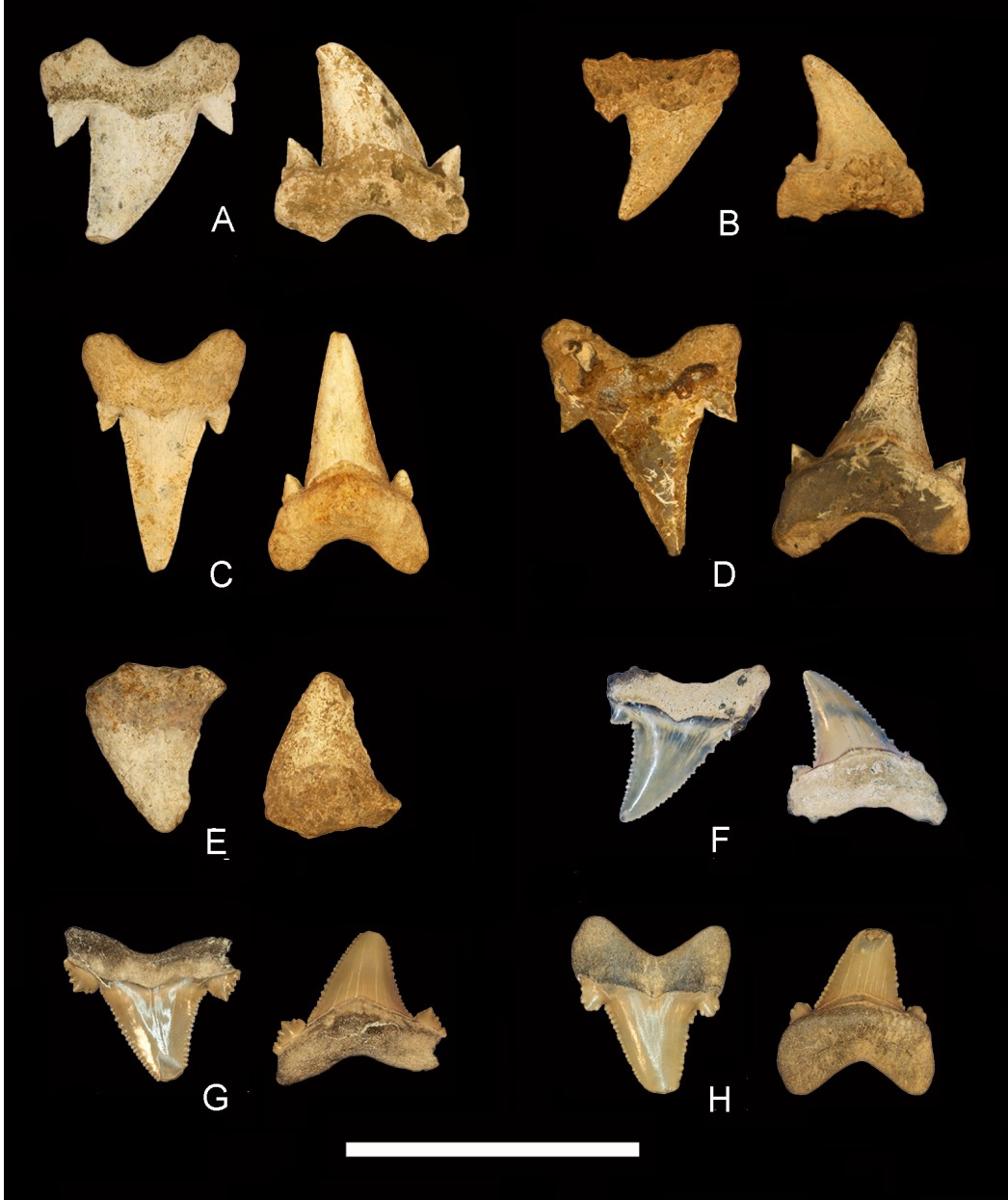

**Figure 3** ***Otodus obliquus*** **and** ***Carcharocles auriculatus*** **teeth from Alabama.** *Otodus obliquus* specimens from Alabama. Labial view on left, lingual view on right. (A) GSA CZ 5051, Unknown formation; (B) GSA CZ 5050, Matthews Landing Marl Mbr.; (C) GSA CZ 5052, Matthews Landing Marl Mbr.; (D) GSA CZ 5053, Porters Creek Fm. (E) GSA CZ 5054, Midway Group; *Carcharocles auriculatus* specimens, labial view on left, lingual view on right. (F) MSC 34423, Lisbon-Tallahatta fms.; (F) ALMNH 1992.28.44.1, Lisbon-Tallahatta fms.; (G) ALMNH 1992.28.44.2, Lisbon-Tallahatta fms. Scale Bar = 5 cm.

of the Matthews Landing Marl, which is the upper member of the Porters Creek Formation (*Smith & Johnson, 1887*). This member falls within the *Morozovella angulata* planktonic foraminiferal zone, placing it within the early Thanetian (*Mancini & Tew, 1988*). The tooth is fairly worn, and it is missing the distal cusplet as well as portions of the root. The main crown is 22.6 mm high and 16.0 mm wide and it does display a prominent v-shaped chevron, making the identification possible.

GSA CZ 5052 (Fig. 3C) was found in the GSA collections with a label inscribed "Naheola, Matthews Landing, Alabama River", referring to the Naheola Formation and the historic locality Matthews Landing which is located along the Alabama River in Wilcox County. The listed formation, however, is likely an error as this locality represents the type section for the Matthews Landing Marl Member, which is the uppermost unit of the Porters Creek Formation. This member underlies the Naheola Formation, but in historical usage, the Matthews Landing Marl was incorrectly thought to be a member of the Naheola Formation (see *Keroher et al., 1966*). Therefore, the specimen can be referred to the very latest Danian or earliest Selandian. GSA CZ 5052 represents an anterior tooth, based on the symmetry of the main cusp. The tooth displays well-developed cusplets, a v-shaped chevron, smooth cutting edges, and a pronounced lingual protuberance of the root. The apex of the crown is chipped, but the remaining portion measures 29.8 mm in height and is 14.3 mm wide.

GSA CZ 5053 (Fig. 3D) is listed as coming from "Grave Yard Hill No. 4" in Wilcox County. Graveyard Hill is another important historic locality in eastern Wilcox County that contains a fossil zone located at the top of the Porters Creek Formation, making the specimen latest Danian or earliest Selandian (*Toulmin, 1977*). The specimen represents an anterior or first posterolateral tooth, with a crown height of 31.0 mm and a crown width of 18.8 mm. GSA CZ 5053 is worn, likely from being exposed for a long period of time, but exhibits a pronounced lingual protuberance, v-shaped chevron, and has well developed cusplets.

The final *O. obliquus* specimen found in collections is GSA CZ 5054 (Fig. 3E). The label associated with this specimen states: "State Collection Midway Group". Although the exact locality for this specimen is unknown, the label indicates that it was discovered within the strata of the Midway Group. This lower to middle Paleocene group includes the Clayton, Porters Creek, and Naheola formations, meaning the tooth is either latest Danian or Selandian. This specimen is highly worn, missing the apex of the crown, cusplets, and most of the root. However, it does preserve the v-shaped chevron, which is diagnostic for the Otodontidae. The remaining portion of the crown is 18.4 mm in height and 19.1 mm wide.

## Remarks

The taxonomic assignment of the Otodontidae is a contentious subject that has been debated for over a century (*Agassiz, 1843*; *Jordan & Hannibal, 1923*; *Glikman, 1964*; *Cappetta, 1987*; *Cappetta, 2012*; *Applegate & Espinosa-Arrubarrena, 1996*; *Zhelezko & Kozlov, 1999*; *Purdy et al., 2001*; *Nyberg, Ciampaglio & Wray, 2006*; *Pimiento et al., 2010*; *Pimiento et al., 2013*; *Ehret, Hubbell & MacFadden, 2009*; *Ehret et al., 2012*). Original descriptions

by *Agassiz (1843)* placed the megatoothed sharks within the Lamnidae, however they have since been reclassified as the Otodontidae by *Glikman (1964)* to recognize their distinct evolutionary history. Since being formally described in the 1840s, the taxonomy of the otodontids has undergone a multitude of changes reflecting reinterpretations of their relationships by a host of researchers (see references above). It is beyond the scope of this study to address the taxonomic stability of the otodontid sharks, however we recognize the genera *Otodus* and *Carcharocles* for the lineage ending with *Carcharocles megalodon*. This arrangement stands in contrast with *Glikman (1964)* and *Cappetta (2012)*, who both referred species with large lateral cusplets to *Otodus*, and those with small or no cusplets to the genus *Megaselachus*. *Cappetta (2012)* revised the taxonomy further, by separating the genus *Otodus* into three subgenera based on the presence, absence, or size of serrations and cusplets as well as differences in root morphology. Furthermore, *Zhelezko & Kozlov (1999)* separated many of the *Otodus* and *Carcharocles* species into subspecies (e.g., *Otodus obliquus mugodzharicus* and *Otodus poseidoni poseidoni*) based on specimens from Kazakhstan. These constructions only further complicate the taxonomy of the Otodontids and do little to elucidate the relationships of the megatoothed sharks. We also argue that, under a biological species concept, it is not possible to recognize subgenera and subspecies in the fossil record. Therefore, we reject these confusing and somewhat subjective designations. Otodontids do likely represent a chronospecific sequence, with individual species derived from a pattern of development that replaces one species with another sequentially through geologic time by incremental morphological and genetic changes (*Applegate & Espinosa-Arrubarrena, 1996*; *Cappetta, 2012*). This mechanism results in a descendant that is much different from its original ancestor, however when looking at smaller time intervals, species distinctions are much more difficult to discern. In the absence of a phylogenetic or a more thorough morphometric analysis, and until further work is conducted and published, we refer the unserrated form to *Otodus obliquus* and serrated forms to the genus *Carcharocles*.

Genus *Carcharocles* Jordan & Hannibal, 1923
*Carcharocles auriculatus* Blainville, 1818
Figs. 3F–3H, 4A–4G, Table 1

### Referred specimens
ALMNH 1985.30.2, ALMNH 1985.35.2, ALMNH 1985.72.28.5, ALMNH 1985.72.33, ALMNH 1985.72.43.3, ALMNH 1985.72.55.2, ALMNH 1985.72.62.3, ALMNH 1985.72.83, ALMNH 1985.72.84, ALMNH 1985.72.88, ALMNH 1988.1.9, ALMNH 1988.29.1, ALMNH 1989.4.50.1, ALMNH 1992.28.44.1, ALMNH 1992.28.44.2, ALMNH 2000.1.4.1, ALMNH 2000.1.16.1, ALMNH 2000.1.27.1, ALMNH 2000.1.29.1, ALMNH 2000.1.33.1, ALMNH 2000.1.53, ALMNH 2000.1.57, ALMNH 2000.1.59, ALMNH 2005.6.259, ALMNH 2005.6.279, ALMNH 2005.6.294, ALMNH 2005.6.334.6, ALMNH 2005.6.407, ALMNH 2005.6.408.1, ALMNH 2010.5.3, MSC 20968, MSC 20969, MSC 20970, MSC 20971, MSC 20972, MSC 20973, MSC 20974, MSC 20975, MSC 20976, MSC

20977, MSC 20978, MSC 20979, MSC 20980, MSC 20981, MSC 20982, MSC 20983, MSC 20984, MSC 20985, MSC 34422, MSC 34423, RMM 2370, RMM 2371.

**Occurrence**

Choctaw, Clarke, Covington, Washington, and Wilcox counties, Alabama

## Description

Characters used to identify *C. auriculatus* in this study include: a large, triangular crown with the presence of lateral serrated cusplets; serrated cutting edges that are fairly coarse and irregular; presence of a v-shaped chevron on the lingual surface of the crown; and developed lingual protuberance on the root. Remains of *Carcharocles auriculatus* are much more prevalent in Alabama than those of *Otodusobliquus*. This difference is likely related to the fact that Middle-Late Eocene deposits are much more expansive in Alabama than are sediments of the Paleocene and Early Eocene.

In the ALMNH collections, 30 specimens of *C. auriculatus* were identified (Figs. 4A, 4E–4G and Table 1). These teeth were collected from Choctaw, Clarke, Covington, Washington, and Wilcox counties in Alabama. Most of the teeth in the ALMNH collections were found in the Yazoo Clay of the Jackson Group and are Priabonian in age. Outcrops of the Yazoo Clay are prevalent throughout the western portion of Alabama and are well known for their marine fossils including early cetaceans such as *Basilosaurus, Zygorhiza,* and *Cynthiacetus* (*Uhen, 2013*). One specimen each of *C. auriculatus* was collected in the Lisbon Formation and the Gosport Sand, which are Lutetian and Bartonian in age, respectively.

Twenty-two *C. auriculatus* specimens were identified in the MSC collections (Figs. 4B–4D, 4H and Table 1). A majority (17) of these teeth were collected from a single locality called Point A Dam in Covington County. Outcrops at this locality represent the boundary of the Tallahatta and Lisbon formations (middle Lutetian; *Clayton, Ciampagalio & Cicimurri, 2013*). The remaining specimens were recovered from the Bartonian Gosport Sand in Choctaw County and the Priabonian Pachuta Marl Member of the Yazoo Clay of Washington County.

One specimen in the collections at MSC (MSC 34423; Fig. 3F) bears resemblance to the Late Paleocene—Early Eocene *Otodus aksuaticus* (*Menner, 1928*). Here, we define *O. aksuaticus* as specimens that exhibit triangular lateral cusplets; a triangular cusp; coarse serrations that fine towards the apex of the cusp; a v-shaped chevron; and a strong lingual protuberance of the root. This species is considered to be part of the chronospecies sequence between *O. obliquus* and *C. auriculatus* (*Zhelezko & Kozlov, 1999*) as it exhibits a transition from the unserrated *O. obliquus* to the serrated *C. auriculatus*, by means of coarse, irregular serrations that do not continue to the apex of the crown. This pattern of serration acquisition is very similar to that seen in the transition from *Carcharodon hastalis* to *Carcharodon carcharias* via *Carchardon hubbelli*, with coarser serrations at the base of the crown, fining towards the apex (*Ehret et al., 2012*).

MSC 34423 is here referred to *C. auriculatus* as it was discovered in sediments located at the Middle Eocene (Lutetian) boundary between the Lisbon and Tallahatta formations in

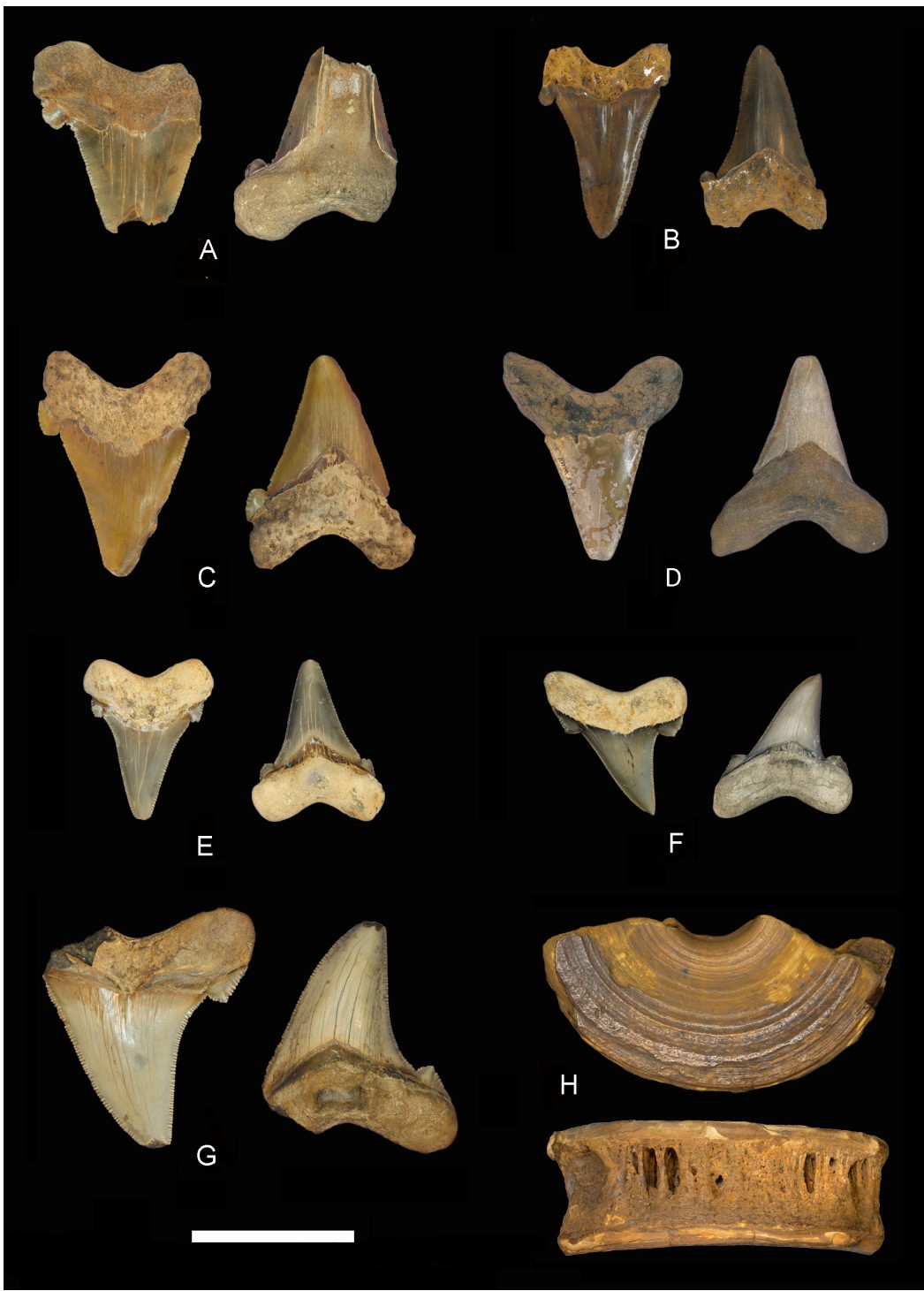

**Figure 4** *Carcharocles auriculatus* **teeth from Alabama.** *Carcharocles auriculatus* specimens from Alabama. Labial view on left, lingual view on right. (A) ALMNH 1988.29.1, Hatchetigbee Fm.; (B) MSC 20970, Lisbon-Tallahatta fms.; (C) MSC 20973, Lisbon-Tallahatta fms.; (D) RMM 2371, Gosport Sand; (E) ALMNH 2000.1.29.1, Yazoo Clay; (F) ALMNH 2000.1.33.1, Yazoo Clay; (G) ALMNH 2005.6.294, Unknown formation; (H) MSC 20968, Lisbon-Tallahatta fms. Scale Bar = 5 cm.

Choctaw County. This specimen does exhibit coarse serrations that fine towards the apex of the cusp and might be close to *O. aksuaticus*, but we refer it here to *C. auriculatus* based on its Lutetian age. MSC 34423 represents a lateral tooth with a crown height of 18.8 mm and a crown width of 14.5 mm. The tooth has an average of 1.2 serrations per mm on both anterior and distal cutting edges.

Two other specimens in the ALMNH collections, ALMNH 1992.28.44.1 and ALMNH 1992.28.44.2 (Figs. 3G–3H), we also refer to *C. auriculatus,* however they demonstrate more coarse serrations that fine towards the apex as seen in the earlier *O. aksuaticus.* The specimens are also Lutetian in age, having been collected at the boundary between the Lisbon and Tallahatta formations in Covington County, Alabama. Both teeth have broken apices and average 1.2 serrations per mm on their cutting edges. Although we assign all three of the aforementioned teeth to *C. auriculatus*, based on their similar morphology, we think there is a good potential for also finding *O. aksuaticus* in Alabama.

One partial vertebral centrum, MSC 20968 (Fig. 4H), recovered from the Point A Dam locality in Covington County, is also referred to *C. auriculatus*. The partial specimen (representing approximately one half of the centrum) is approximately 11.2 cm in diameter and 3.9 cm in thickness. The centrum is typically lamniform in appearance, and is laterally compressed with concave articular surfaces and radiating calcified lamellae within the intermedialia. Only one pit is preserved for the insertion of either the neural or haemal arch, however the centrum is fragmentary and it cannot be deduced as whether or not it is dorsal or ventral. We are confident in referring this specimen to *C. auriculatus* because of its lamniform appearance, age, and large size. Other lamniform taxa recovered from the Point A Dam locality are primarily odontaspids (*Clayton, Ciampagalio & Cicimurri, 2013*), which would not have centra this large. For example, *Hansen et al. (2013)* reported a 6th vertebral centrum diameter of 30 mm for a recent *Odontaspis ferox* specimen with a total body length of 297 cm, which is significantly smaller than our fossil specimen.

## DISCUSSION

Surprisingly, the presence of otodontid sharks in Alabama has not been extensively reported in the literature. *Agassiz (1843)* noted the presence of *Otodus crassa* in Alabama, although he provided no additional details. *Leriche (1926)* synonymized *O. crassa* with *Carcharodon hastalis*, which is probably correct for some of the specimens figured in *Agassiz (1843)*. However, the Miocene *C. hastalis* has not been reported from Alabama and at least one of the specimens figured in *Agassiz (1843)* appears to be *O. obliquus*. As a result, it stands to reason that *Agassiz (1843)* might have been the first researcher to identify *O. obliquus* from Alabama. A few years later, *Gibbes (1848)* described the presence of *Otodus crassus* within the Cretaceous of Alabama. Describing what appears to be *Carcharocles auriculatus,* Gibbes, like many researchers at the time, mistakenly referred the Eocene deposits in the state to the Upper Cretaceous (*Ebersole & Dean, 2013*). Since that time, however, no other *Otodus* teeth are known to have been reported in Alabama. As discussed above, many of the *Otodus* teeth in the GSA collections were misidentified as *Odontaspis*, *Lamna*, or *Carcharias*, which could have confounded the

situation. Furthermore, outcrops of the Midway and Wilcox groups are not widely exposed in Alabama, making it difficult to find Paleocene and Early Eocene fossils. Another large *O. obliquus* specimen was observed by one of the authors in the collection of a private collector; unfortunately this specimen could not be secured for the ALMNH collections at the present time (D Ehret, pers. obs., 2013). *Otodus* specimens have also been found in nearby states including the Williamsburg Formation of South Carolina (*Purdy, 1998*) and the Tuscahoma Formation of Mississippi (*Case, 1994*). These discoveries leads us to propose that *Otodus* teeth might be more prevalent in Alabama than previously thought if the proper aged outcrops are targeted for collecting.

*Carcharocles auriculatus* is the megatoothed species that is more commonly found in Alabama. Its predominance is likely a result of the comparably numerous outcrops of the Eocene Tallahatta and Lisbon formations and the Yazoo Clay. The large size of the teeth is also likely a factor in their discovery and collection. The closely related *Carcharocles angustidens* was reported from Alabama by *White (1956)* and *Thurmond & Jones (1981)*. The specimens discussed in *White (1956)* that are housed in the British Museum (NHM London) are referred to the Jackson Group and, based on specimens discussed and figured here, are most likely *C. auriculatus*. *Thurmond & Jones (1981)* figured a specimen (Fig. 22, pg. 56) referred to *C. angustidens* as a line drawing, which was reported as being part of the former Birmingham Southern College collections and collected from an unknown locality. Unfortunately the whereabouts of this specimen are unknown, and the poor quality of the figure does not allow for an accurate identification. *C. angustidens* is a species of otodontid recorded from the Oligocene that exhibits a larger overall tooth size, smaller cusplets, and finer serrations than *C. auriculatus*. Because there are relatively few Oligocene outcrops in Alabama and the relatively high prevalence of Eocene outcrops, we are confident that all records of *Carcharocles* in Alabama thus far represent *C. auriculatus*. Furthermore, prospecting Oligocene sediments in Alabama by both authors has only yielded small to microscopic chondrichthyan teeth of Carcharhiniformes, Ginglymostomatidae, and Myliobatidae. No occurrences of *Carcharocles chubutensis* or *Carcharocles megalodon* have been accurately reported from Alabama, likely a result of the historic lack of systematic collecting in the Mio-Pliocene formations in the state. However more concentrated collecting efforts in southern Alabama where Oligocene-Pleistocene deposits are more concentrated may yield new specimens. Additionally, the use of historic collections (e.g., Geological Survey of Alabama collections) can be a valuable resource in identifying overlooked or misidentified specimens.

## CONCLUSIONS

The fossil record of otodontid sharks in Alabama has gone largely unreported in the literature. Reviews of the collections at the Alabama Museum of Natural History, McWane Science Center, and the Geological Survey of Alabama have yielded late Paleocene through Eocene otodontids including *O. obliquus* and *C. auriculatus* from the state. This study represents the first reliable report of *Otodus* from Alabama, with specimens identified from multiple localities. *Otodus obliquus* was identified in the collections at the Geological

Survey of Alabama, most of which were collected prior to 1910. Based on observations of amateur collections, we think that the presence of *O. obliquus* is likely more common than what the few specimens in museum collections suggest. *C. auriculatus* is the most common otodontid shark found in Alabama, typically recovered from Lutetian-Ypresian outcrops in southwestern Alabama. While large specimens are not as common as they were 50–100 years ago, teeth assigned to this taxon are still recovered with some regularity. We also refute *Thurmond & Jones*' (*1981*) report of *C. angustidens* from the state. This specimen was most likely *C. auriculatus*, however, its status is unknown until the tooth can be rediscovered. *C. chubutensis* and *C. megalodon* are currently not known from Alabama, but with increased collection in the southern Cenozoic deposits in the state, specimens might be recovered.

## ACKNOWLEDGEMENTS

Many thanks to Sandy Ebersole, Geological Survey of Alabama, for access to the fossil collections and help compiling the geologic data for Fig. 1. The authors would also like to thank Dave Cicimurri, South Carolina Museum of Natural History, and Chuck Ciampaglio, Wright State University, for useful discussions on otodontid records. We would also like to thank the reviewers of this manuscript.

### Funding

The University of Alabama Museums (Tuscaloosa, AL) and the McWane Science Center (Birmingham, AL) provided funding for this project. The funders had no role in study design, data collection and analysis, decision to publish, or preparation of the manuscript.

### Grant Disclosures

The following grant information was disclosed by the authors:
The University of Alabama Museums.
McWane Science Center.

### Competing Interests

The authors declare there are no competing interests. Jun Ebersole works for McWane Science Center, a non-profit 501(c)(3) organization.

### Author Contributions

- Dana J. Ehret conceived and designed the experiments, performed the experiments, analyzed the data, contributed reagents/materials/analysis tools, wrote the paper, prepared figures and/or tables, reviewed drafts of the paper, visited private collections.
- Jun Ebersole conceived and designed the experiments, performed the experiments, analyzed the data, contributed reagents/materials/analysis tools, wrote the paper, prepared figures and/or tables, reviewed drafts of the paper, conducted the geologic review of Alabama and the historic review of paleontology collections.

**Peer**J

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
