# Peer review of "Occurrence of the megatoothed sharks (Lamniformes: Otodontidae) in Alabama, USA"

_PeerJ, doi:10.7717/peerj.625_

## Round 0.1 · original submission · Minor Revisions

I agree with all of the reviewers, who found the manuscript worth publishing, but with minor revisions. Descriptive paleontology papers ought to be more common in PeerJ, and manuscripts like this one help pave the way. Please pay attention to the line by line edits from all of the reviewers because there is a need to improve the mechanics of the text. Reviewer 1's concern about the focus on C. megalodon bookending the manuscript is not really pertinent in my view -- they are appropriate hooks for non-specialist readers, and there is no need to delve into the matter for this paper. Reviewer 2’s comments in particular should be considered in depth, especially with respect to the need to tie in figure mentions with the manuscript text (an item identified by Reviewer 3 as well). None of these changes is onerous, and collectively they will only improve the readability of the paper. Some additional necessary edits:

Lines 11, 199, 264. Uncapitalize megatoothed and odontaspids.
Line 18. Delete semi-colon or edit to present a list with a colon and separate items by semi-colon.
Lines 48-50. Suggested edit “…lower-most Paleocene age Clayton Formation.”
Line 155. Correct as “…their marine fossils including…”
Line 156. Could clarify which “one specimen” refers to here: C. auriculatus or cetacean?
Line 187. Space is not an issue; please spell out Alabama.
Lines 96, 190, 221 & 251. Diction: change “believe” to “think”, “argue”, or “propose”
Lines 250-251. A bit ungainly to speak in the third person here with “…both authors…” Instead, just say “we think…”

Please control for font changes within figures and across figures. I propose keeping Arial from Figure 1 (but adjust abbreviations and unit scales); adjust font in the color keys in Figure 2 to match; and similar consistency with Figures 3-4. Variable fonts are distracting.

·

Basic reporting

No comments. The article meets the standards.

Experimental design

No comments. The article meets the standards.

Validity of the findings

No comments. The article meets the standards, taking into consideration it is a systematic paleontology paper, some of the criteria do not apply. However, the findings are valid.

Additional comments

This paper reports the occurrence of the megatoothed sharks Otodus obliquus and Carcharocles auriculatus in the Paleocene and Eocene of Alabama. Because the fossil record of megatoothed sharks from this area is out to date, this study contributes to the field of vertebrate paleontology. Furthermore, given that records of megatoothed sharks are rare in this area, this study could potentially provide the basis for future research on this group. Finally, this study is an example of the use of historic museum collections to advance the field of vertebrate paleontology. This subject has great potential not only for research, but also for education and outreach; hence, it should be highlighted and promoted. That being said, I think this paper should be publish in PeerJ, after minor changes are made.

General comments:

The authors mention other megatoothed species -C. megalodon- in the abstract, introduction and the conclusions. However, it is not clear how the work presented is relevant for the knowledge of this particular species. Although I recognize it is a charismatic species that receives a great deal of attention, I cannot see how the present work, as it is, will advance the understanding of C. megalodon, nor why that is important. It could be argued that a more effective way to improve the fossil record of C. megalodon is by focussing in Miocene deposits and not in the finds reported here. I would suggest to either to elaborate more on this idea, or to remove that part. Otherwise, the C. megalodon lines may look like “throw-away” lines.

The systematic paleontology section should be presented in a more structured way so it is easier to the reader to interpret the results.
For instance, it would be best if this section begins with a list of specimens described after the subheading: “Specimens studied”. There, the number of specimens and the accessory numbers should be provided.
An “Occurrence” subheading should go next, with the information of where the specimens where collected from. This section should be tied up to the “Geological setting” part of the methods, and to the Figure 1.
The “Description” section should be next, starting with a *general* description of the diagnostic characters of the specimens. Also, it should include those characters that separate that taxa from other relevant species. Then, a more specific description of the different specimens should be take place.
Finally, a “Discussion or Remarks” subheading should be next, with any other considerations, including the implications of the occurrences and other notes on the age, distribution, etc.
These suggestions are merely structural, but I think they will make a difference, and hopefully, they will make the article easier to follow, specially for non-paleontologists that may be interested in only parts of the systematic paleontology.

Finally, I think that the authors could highlight some more their use of historic museum collections, and provide some recommendations about their importance and potential.

Specific comments:

Lines 14-16: This would be a better place to discuss how megatoothed sharks are a series of chronospecies. Also, to add the broader implications of this work for the understanding of apex predatory sharks.

Lines 18-20: How this study advances the knowledge of the differences in abundance of megatoothed species?

Line 21: Not sure if accuracy is the appropriate word here. Perhaps, out to date?

Lines 30-35: That should be part of the introduction, not the methods.

Lines 36-42: You mention the number of specimens from the GSA, but not from ALMNH and MSC.

Line 44: Start this section by explicitly saying from what geologic units the specimens from this study were collected from.

Lines 64-65: Any general information on the collection methods? Any information on the environment of deposition of these sites?

Lines 77-78: Also discussed in: Pimiento et al. (2013) Early Miocene chondrichthyans from the Culebra Formation, Panama: A window into marine vertebrate faunas before closure of the Central American Seaway. Journal of South American Earth Sciences 42: 159-170 http://dx.doi.org/10.1016/j.jsames.2012.11.005.

Lines 80-81: Not relevant for this work.

Lines 95-97: That should not be part of the description section.

Lines 41-200: Please take into consideration the comments above, as well as the general comments.

Lines 240-242: This would be a good place to elaborate some more on the importance of historic collections: How they might be overlooked while holding a great deal of potential for the advance of the field of vertebrate paleontology.

Lines 243-259: This would be a good place to provide recommendations for future works, and the use of historic museum collections.

·

Basic reporting

As far as I can discern the article matches all of the PeerJ requirements for basic reporting. The manuscript uses a couple of departures from the standard "results" section, instead replacing this section with a "Systematic Paleontology" section (and subdivided into a taxonomic remarks section and descriptions) - this is a pretty standard convention in paleontology, and is in this case an expected modification. Otherwise, the manuscript is written in clear English, and includes a concise and well-written introduction. The figures are of excellent quality. However, figures 1 and 2 include a geologic map and stratigraphic chart of Alabama (respectively) but no citations are given in the figure captions or in the Materials and Methods section. This is almost certainly a simple oversight, and citations for the source of the map and stratigraphic units, foraminiferal zones, and nannoplankton zones are necessary and ought to be included.

Experimental design

The purpose of the paper is clearly stated at the end of the introduction as reviewing the available fossil record of megatoothed sharks from the Paleogene of Alabama. These authors scoured fossil collections at three different natural history museums in Alabama, relying upon specimens within permanent collections (e.g. rather than private collections). Surveying collections for specimens such as described in the Material and Methods section is standard in paleontology.

The authors refer at one point to a specimen in a private collection - I think that this is acceptable because the authors only make note of it, and do not describe the specimen or base any conclusion upon it.

Validity of the findings

These authors report many new occurrences of Otodus and Carcharocles, and make several necessary corrections to prior misidentifications of these sharks from the Alabama fossil record. The Paleogene is critical to understanding the Otodus-Carcharocles transition, which has been studied in some detail in various localities in Asia, Africa, and Europe, but as acknowledged by the authors, Paleogene records of these genera in North America are rare. A series of chronospecies have been named as transitional species between Otodus obliquus and Carcharocles auriculatus, but the original publication is in Russian and as of yet has not been translated, which is a bit of a stumbling block for western authors. Clarification of the timing of occurrence of Otodus obliquus and Carcharocles auriculatus in well-dated strata in the Gulf and Atlantic Coastal Plain offers serious advances towards understanding the exact chronology of morphological changes in the otodontid lineage. The conclusions of the paper are indeed appropriately stated and directly address the research question. Lastly, the occurrence data for otodontid specimens discussed in the paper are included within Table 1. Speaking of, table 1 could be further improved by including an additional column listing the geologic age of each occurrence (e.g. Priabonian, late Paleocene).

Additional comments

I have some recommended edits listed here. Listed first are edits I think are necessary, followed by edits that are suggestions which I feel would improve the paper.

Necessary edits

-Citations to Figure 3 and 4 - in the text, these are simply given as "Fig. 3". For citations to individual elements of the figure, the letters should be given as well (e.g. Fig 4D-E) so the reader does not have to flip back and forth between the text and figure caption to make sure they're looking at the correct part of the figure.

-Introduction, line 17: change "other Otodus..." to "other species of Otodus and Carcharocles..."

-Geologic setting, line 47: change to K/Pg boundary.

-Also line 47: "contact" would be better replaced with "boundary."

-Remarks section in Syst. Paleo - some additional background is probably necessary. The text refers to subgenera without citing Cappetta (2012) who made some admittedly bizarre changes to otodontid taxonomy including the use of subgenera for otodontids. Otodontids are widely thought to be a single anagenetic lineage through most of the Cenozoic, and any "vertical" subdivision of the "ladder" will result in a series of temporally separated chronospecies. This is widely acknowledged, and Cappetta's (2012) taxonomic changes will only serve to further muddy the waters rather than clarify what we know about the megatoothed sharks. A few extra sentences explaining various taxonomic proposals/paradigms (e.g. Jordan and Hannibal 1923, Glikman, Cappetta, Purdy et al., Gottfried et al., etc.) are necessary in order for the statement about subgenera and subspecies to make sense. Also, some statement indicating that subgenera are not really a taxonomic convention in vertebrate zoology would be sufficient.

-Description, Line 188: A word is missing here. Could be fixed by changing to "Although we assign..."

-Description, line 194: change "typical" to "typically"

-Discussion, line 216: what do the authors mean by "abundant"? This is perhaps vague when applied to lithostratigraphic units. If the authors wish to indicate that the unit is not widely exposed or is very thin and unfossiliferous (or both), then a more descriptive term is appropriate.

-Discussion, line 217: "impressive" is a bit vague, and should be replaced with a better descriptor like large or well-preserved.

-Discussion, lines 217-219: a single personal observation citation (rather than two) is sufficient here if changed to "...private collector; unfortunately, this specimen could not be secured for the ALMNH collections at the present time (D. Ehret, pers. observ.).

-Discussion, line 229: "...now defunct Birmingham Southern..." - perhaps "former" might better replace "now defunct"; also, I'm guessing that "Birmingham Southern" refers to Birmingham Southern College; the full institution name should be included here (as the institution is not listed in the materials and methods section)

-Discussion, line 230: I think "whereabouts" is plural, so "is" should be changed to "are". I looked this up as "whereabouts is" doesn't sound right, and apparently both are grammatically correct but a poll of English professors found that a majority preferred "whereabouts are". I'll leave this one to the editor.

-Discussion, line 235: change to "thus far represent C. auriculatus"

-Also line 235: "collecting of Oligocene sediments" is awkward unless the authors intend that the sediments themselves are being collected; I'm assuming the authors intended prospecting or collecting of shark teeth from those sediments, and if this is the case, it could be changed to "Furthermore, prospecting of Oligocene sediments" or "Furthermore, field study of Oligocene sediments" or something similar.

-Discussion, Line 237: "Furthermore" is repeated here as the first word in two consecutive sentences.

-Discussion, line 239: because the sentence beginning on line 237 is not very long, these two sentences could be joined as such: "...reported from Alabama, likely a result of the historic lack of systematic collecting..."

Suggested edits

-I'm not sure if the authors have further plans past this manuscript, but the big elephant in the room is the question of central Asian transitional species (e.g. O. obliquus var. mugodzharicus, C. poseidoni, C. aksuaticus, C. sokolovi) and how they fit in. The authors acknowledge that one tooth in particular shares apically diminishing serrations like C. aksuaticus, but refer it to C. auriculatus based on its age. It may be beyond the scope of the paper, but I think some background on the transition - and even some commentary on the use of these transitional species - would be appropriate and certainly welcome. At the very least, the data presented in this paper give robust times for O. obliquus and C. auriculatus in North America, which would bracket some of these transitional species (and I personally think the significance of this is downplayed in the article). Also, an expanded discussion of this topic could provide some rationale for the referral of the C. aksuaticus-like tooth to C. auriculatus (e.g., juvenile teeth of otodontids often appear like the adults of the ancestors; juvenile C. megalodon teeth will resemble adult C. angustidens or C. chubutensis teeth, for example).

-Description, line 151: "referred" and "identified" are used in the same sentence here; it would be sufficient to use one - e.g. 30 specimens of C. auriculatus were identified, or we referred 30 specimens to C. auriculatus.

-Description, line 184: "demonstrate more coarse" could be replaced with "possess coarser"

-Description, lines 187-188: this sentence could be shortened by changing it to: "Both teeth have broken apices and average 1.2 serrations per mm on their cutting edges."

-Description, line 199: "Odontaspids" should be in lower case.

-Lines 199-200: Could the authors provide a "ballpark" estimate of how large odontaspid vertebral centra would measure, based upon extant Odontaspis?

-Discussion, line 236: "turned up" ought to be replaced with "yielded"

-Conclusions, line 259: May want to change Tertiary to Cenozoic.

·

Basic reporting

"No Comments"

Experimental design

"No Comments"

Validity of the findings

"No Comments"

Additional comments

All finding are valid, and presented very clearly. This article is succinct, clear, and self-contained. It brings to light an important aspect of elasmobranch distribution and evolution, concentrating on Paleocene and Eocene mega-toothed sharks. In showing that both Otodus and Carcharocles are found in some abundance in Alabama sediments, this expands upon their cosmopolitan nature, and evolutionary trends.

The authors should clearly state that they are defining breaks in chrono-species via stratigraphic/temporal units. It is suggested in the paper, but perhaps a few sentences devoted to this would strengthen the paper.

The authors provide an excellent figure, Figure 4", but fail to reference nearly all of this figure in the body of the paper. A brief mention to the specimens pictured would be useful.

The figure caption for Figure 3 should be carefully checked for correct reference to each specimen pictured. There seems to be an incongruous lettering of images.

---

## Round 0.2 · Minor Revisions

Dear Dr. Ehret,

Thank you for your revisions (version 1) on your manuscript. The changes to the main document have addressed all of the crucial concerns and suggestions of the reviewers, and the manuscript should be acceptable for publication pending one last final set of revisions. Hence, this decision constitutes as a minor revision, with the needed changes enumerated below.

The additional edits include some suggested changes that should be made to the typography and diction, and these are highlighted in the Track Changes of the edited document, which will be forwarded to you by the PeerJ Editorial Team. (Currently there is no mechanism to attach such a document, but the system will be upgraded soon to permit this process).

Some important changes also need to be made to the figures and figure captions:

- The font for the labels on the geologic map and the legend in Figure 1 should be in Arial, for consistency with the rest of the manuscript. The scale bars should have a numeric value and unit either labeled or identified in the caption. Version 0 was fine in this regard.

- Figures 1, 3-4 are missing full captions (which should be added to their existing titles); they were present in the original submission (version 0), and they could be easily be restored in the the next version without any editing. I presume their absence is a simple slip of the keystroke.

---

## Round 0.3 · Minor Revisions

This manuscript is 99% ready for acceptance; I appreciate the authors' diligence and consideration regarding the small edits to the text and figures.

The only remaining item:

The scale bar on the bottom left of Figure 4 shows a thick line (presumably the scale bar) and a fainter thin line below it (a vestige of formatting and scaling?). Please delete the latter, if so. There should only be 1 scale bar, as in Figure 3.

---

## Round 0.4 · accepted · Accept

This manuscript is ready to be published!